# Diazoxide Needs Mitochondrial Connexin43 to Exert Its Cytoprotective Effect in a Cellular Model of CoCl_2_-Induced Hypoxia

**DOI:** 10.3390/ijms222111599

**Published:** 2021-10-27

**Authors:** Michela Pecoraro, Stefania Marzocco, Ada Popolo

**Affiliations:** Department of Pharmacy, University of Salerno, 84084 Fisciano, Italy; mipecoraro@unisa.it (M.P.); smarzocco@unisa.it (S.M.)

**Keywords:** mCx43, K_ATP_ channels, diazoxide, chemical hypoxia

## Abstract

Hypoxia is the leading cause of death in cardiomyocytes. Cells respond to oxygen deprivation by activating cytoprotective programs, such as mitochondrial connexin43 (mCx43) overexpression and the opening of mitochondrial K_ATP_ channels, aimed to reduce mitochondrial dysfunction. In this study we used an in vitro model of CoCl_2_-induced hypoxia to demonstrate that mCx43 and K_ATP_ channels cooperate to induce cytoprotection. CoCl_2_ administration induces apoptosis in H9c2 cells by increasing mitochondrial ROS production, intracellular and mitochondrial calcium overload and by inducing mitochondrial membrane depolarization. Diazoxide, an opener of K_ATP_ channels, reduces all these deleterious effects of CoCl_2_ only in the presence of mCx43. In fact, our results demonstrate that in the presence of radicicol, an inhibitor of Cx43 translocation to mitochondria, the cytoprotective effects of diazoxide disappear. In conclusion, these data confirm that there exists a close functional link between mCx43 and K_ATP_ channels.

## 1. Introduction

Hypoxia is the leading cause of myocardial cell damage, characterized by apoptotic cell death ascribed to an imbalance between the level of reactive oxygen species (ROS) and the antioxidant defence system. In the heart, chronic hypoxia and the consequent increased levels of hypoxia-inducible factor 1 (HIF 1), alter Ca^2+^ handling, change the expression of gap junction proteins and induce angiotensin II production, thus leading to atrial fibrosis and structural remodelling [1]. However, cells respond to hypoxia by complex metabolic reprogramming and molecular mechanisms aimed to minimize the detrimental consequences of oxygen deprivation on mitochondria [2]. Cytoprotective programs activated during hypoxia injury integrate several processes, including the hypoxia-inducible factor 1/hypoxia response element pathway [3], translocation of connexin43 (Cx43) to the inner mitochondrial membrane [4] and opening of the mitochondrial ATP-regulated potassium (mitoK_ATP_) channels [5].

Cx43 is a phosphoprotein expressed both on the plasma membrane and on mitochondria. Transmembrane Cx43 contributes to cell–cell communication and electrical coupling by the formation of gap junction channels, while on the mitochondrial membrane Cx43 exists as a hemichannel and is involved in mitochondrial volume regulation and respiration [6]. During stress conditions, Cx43 translocates to the mitochondria with a mechanism that involves the Hsp90/Tom20 machinery system [4,7], and many studies conducted using the Hsp90 inhibitor radicicol supported this theory [8,9,10]. Mitochondrial Cx43 (mCx43) acts as an important regulator of apoptosis, allowing the passage of molecules that induce apoptosis such as Ca^2+^, IP_3_ and cAMP ions [11] and influencing mitochondrial respiration, matrix ion fluxes and ROS production [9]. Indeed, mitochondria are at the same time the most powerful intracellular source and the primary target for damaging effects of ROS. Furthermore, mitochondrial ROS overproduction facilitates the calcium dependent mitochondrial permeability transition, which plays a key role in caspase-mediated apoptosis [12]. The inhibition of Cx43 translocation on mitochondria induces a drastic increase in ROS production, interferes with calcium homeostasis and accelerates mitochondrial membrane depolarization, thus inducing apoptosis [7,8,9,13].

MitoK_ATP_ channels mediate the electrophoretic uptake of potassium driven by the negative mitochondrial membrane potential [14]. These channels are activated by pharmacological agent such as diazoxide (dzx), whereas glybenclamide (gly) and ATP inhibit mitoK_ATP_ channel activity [15]. Under conditions of metabolic stress, the activation of K_ATP_ channels is an important part of endogenous protective signalling that promotes cellular survival [16]. The opening of mitoK_ATP_ channels reduces the expression of apoptotic markers induced by the ischemia/reperfusion injury [17,18], while the inverse occurs when the same channels are closed [16]. A previous study reports that the pharmacological protection exerted by dzx is abolished when Cx43 translocation on mitochondria is inhibited [4], however, the functional link between mCx43 and K_ATP_ channels is not fully established.

Mitochondrial Cx43 and K_ATP_ channels are part of the same signalling pathway activated to confer cell protection against ischemia/hypoxia conditions [16], and this study aimed to demonstrate that they act in a functionally dependent way to reduce cellular damage in an in vitro model of chemical hypoxia-induced apoptosis. For this purpose, we used cobalt chloride (CoCl_2_) to induce chemical hypoxia in H9c2 cells as previously reported [19]. Radicicol was used to inhibit Cx43 translocation to mitochondria in order to evaluate solely the contribution of K_ATP_ channels to cardioprotection. Dzx was used to open K_ATP_ channels and thus evaluate the additive cardioprotective effects of K_ATP_ channels and mCx43, whereasgly was used to close K_ATP_ channels and thus evaluate the effects of mCx43 alone.

## 2. Results

### 2.1. The Effects of Diazoxide on Mitochondrial Cx43 Expression in CoCl_2_-Treated H9c2 Cells

In order to evaluate the expression levels of Cx43 and its phosphorylated form (mpCx43) on mitochondria, we performed a Western blot on mitochondrial cell lysates, and we used Cx43 antibody in agreement with the manufacturing instructions. As previously reported [20,21], two bands appear on the blot; the lower one corresponds to Cx43 and the upper one to Cx43 phosphorylated on Ser 368 (pCx43). Cx43 expression was normalized to Tom20, used as a loading control. As reported in panel C, the Western blot analysis showed a significant increase (*p* < 0.05) in mCx43 expression in CoCl_2_-treated H9c2 cells. In radicicol pre-treated cells, no significant differences compared to untreated cells were observed, while a significant (*p* < 0.05) reduction of mCx43 levels compared to CoCl_2_-treated cells was observed. The closure of K_ATP_ channels exerted by gly induced a significant (*p* < 0.05) increase of mCx43 expression compared to untreated cells. The activation of K_ATP_ channels exerted by dzx pre-treatment induced a significant (*p* < 0.005 vs. untreated cells) increase in mCx43 expression. Radicicol reverted this effect; indeed in dzx and radicicol co-treated cells, mCx43 expression was significantly (*p* < 0.05) lower than in CoCl_2_ and dzx co-treated cells (Figure 1C).

The mpCx43 expression was normalized to Cx43 in order to evaluate the relationship between the expression levels of the two protein isoforms. The mpCx43/mCx43 ratio was significantly (*p* < 0.001) higher in CoCl_2_-treated cells as well as in dzx pre-treated cells compared to control cells. In radicicol and radicicol and dzx co-treated cells, the mpCx43/mCx43 ratio was significantly (*p* < 0.001) lower than in both the CoCl_2_-treated cells and CoCl_2_ and dzx co-treated cells (Figure 1D).

### 2.2. Mitochondrial Cx43 and Mitochondrial K_ATP_ Channels Are Involved in CoCl_2_-Induced Superoxide Production

The mitochondrial superoxide production, evaluated using MitoSOX red, was significantly (*p* < 0.05) higher in CoCl_2_-treated cells compared to untreated cells. In radicicol pre-treated cells, the mitochondrial ROS production was significantly (*p* < 0.001) higher than in untreated cells. In dzx pre-treated cells, no significant difference in mitochondrial ROS production compared to untreated cells was observed, whereas a significant (*p* < 0.05) reduction compared to CoCl_2_-treated cells appears. The closure of K_ATP_ channels exerted by gly administration significantly (*p* < 0.001 vs. untreated cells) affects mitochondrial superoxide production. The inhibition of Cx43 translocation to mitochondria exerted by radicicol drastically reverts the effects of dzx. Indeed, in radicicol and dzx co-treated cells mitochondrial superoxide production was significantly (*p* < 0.001) higher than in CoCl_2_-treated cells and also significantly (*p* < 0.001) higher than in CoCl_2_ and dzx co-treated cells (Figure 2).

### 2.3. Mitochondrial Cx43 and the Activation of Mitochondrial K_ATP_ Channels Prevent Mitochondrial Membrane Depolarization

In order to evaluate mitochondrial membrane depolarization, we used the cationic dye TMRE, which is trapped in mitochondria and emits fluorescence in inverse proportion to mitochondrial membrane potential. As reported in Figure 3, CoCl_2_ significantly (*p* < 0.001 vs. untreated cells) induced mitochondrial membrane depolarization; indeed, the percentage of TMRE positive cells is lower than control cells. Additionally, in radicicol-treated cells, as well as in gly-treated cells, a significant (*p* < 0.005 and *p* < 0.05 vs. untreated cells) mitochondrial membrane depolarization was observed. Following the activation of mitochondrial K_ATP_ channels through dzx pre-treatment a significant (*p* < 0.005) improvement in mitochondrial membrane potential was observed compared to the cells treated with CoCl_2_ alone (Figure 3).

### 2.4. Mitochondrial Cx43 and the Activation of Mitochondrial K_ATP_ Channels Reduce CoCl_2_-Induced Ca^2+^ Homeostasis Dysregulation

Mitochondrial calcium content plays a pivotal role in apoptosis induction, so we evaluated the percentage of delta increase in the intracellular calcium concentration induced by CoCl_2_, both in the absence and presence of pharmacological tools. CoCl_2_ significantly (*p* < 0.005 vs. untreated cells) increased the cytosolic calcium content; indeed, the higher delta increase indicates that higher levels of calcium were stored in intracellular organelles. Radicicol pre-treatment significantly (*p* < 0.05) increased the effect of CoCl_2_ and the percentage increase in delta was also significantly (*p* < 0.001) higher than in CoCl_2_ and dzx co-treated cells. In dzx pre-treated cells no significant difference in intracellular calcium content was observed compared to control cells, whereas a significant (*p* < 0.05 vs. CoCl_2_-treated cells) reduction in the CoCl_2_ induced effect was observed; in fact, the percentage increase in delta in the intracellular calcium concentration was lower than in CoCl_2_-treated cells (Figure 4A). In the same way we observed that the percentage increase in delta in [Ca^2+^]_i_ induced by FCCP, a mitochondrial Ca^2+^ depletory, in CoCl_2_-treated H9c2 cells as well as in radicicol-treated cells was significantly (*p* < 0.001) higher than that found in untreated cells, indicating higher levels of Ca^2+^ stored in mitochondria. Both in dzx- and gly-treated cells the percentage increase in delta induced by FCCP was significantly (*p* < 0.05) lower than in CoCl_2_-treated cells, indicating lower levels of Ca^2+^ stored in mitochondria. In rad and dzx co-treated cells the percentage increase in delta was significantly (*p* < 0.05) lower than in both CoCl_2_-treated and CoCl_2_ and dzx co-treated cells (Figure 4B). 

### 2.5. Mitochondrial Cx43 and the Activation of Mitochondrial K_ATP_ Channels Counteract CoCl_2_-Induced Apoptotic Response

Myocardial hypoxia is a main cause of cardiac dysfunction due to its triggering apoptosis. Cytofluorimetric analysis by means of propidium iodide showed that CoCl_2_ leads to a significant (*p* < 0.005) increase in the percentage of hypodiploid nuclei, and the inhibition of Cx43 translocation to mitochondria exerted by radicicol pre-treatment further increases the deleterious effects of CoCl_2_ (*p* < 0.001 vs. CoCl_2_ alone). The inhibition of mitochondrial K_ATP_ channels exerted by gly does not influence the effects of CoCl_2_ on the apoptosis induction, since no significant difference in the percentage of hypodiploid nuclei compared to the cells treated with CoCl_2_ alone was observed. On the contrary, the activation of mitochondrial K_ATP_ channels exerted by dzx significantly (*p* < 0.005) reduced the percentage of hypodiploid nuclei, as compared with CoCl_2_-treated cells. The contemporary administration of rad and dzx reverted the beneficial effects of dzx, since in cells treated with both drugs, a significant (*p* < 0.05) difference in the percentage of hypodiploid nuclei with respect to CoCl_2_ and dzx co-treated cells was observed (Figure 5).

## 3. Discussion

Hypoxia is a cellular injury characterized by brief or sustained episodes of oxygen deprivation. A limited supply of oxygen or damage in oxygen processing activates signalling pathways that result in structural and functional changes involved in the adaptation of myocardium to pathological conditions [22]. Notably, CoCl_2_ is commonly used to induce hypoxic environments by replacing Fe^2+^ in haemoglobin to form deoxygenated haemoglobin. CoCl_2_ has been used for hypoxic preconditioning in many cellular models and has been found to induce cellular damage, decrease mitochondrial membrane potential, and induce apoptosis in many cell types [19]. In our previous study we showed that CoCl_2_ administration increases mitochondrial ROS production and affects mitochondrial membrane potential and intracellular Ca^2+^ homeostasis, thus inducing apoptosis in H9c2 cells. Furthermore, the inhibition of Cx43 translocation to mitochondria further increases the apoptotic effects of CoCl_2_-induced hypoxia [13].

Cx43 is a member of the connexin family, widely expressed in many cellular types, that mediates intercellular communication by promoting the passage of small molecules and ions. In cancer cells, Cx43 is responsible for the propagation of apoptotic signals, while in cardiomyocytes Cx43 regulates electrical coupling and synchronous contraction [23], and both mechanisms are regulated by the flux of Ca^2+^ through the Cx43-formed gap junction [11]. In cardiomyocytes, Cx43 is also expressed on mitochondrial membranes [24] where, as a hemichannel [25], it takes part in cardioprotection [26]. Indeed, the activity of Cx43 in cardiomyocytes is necessary for the transport of potassium ions from the cytosol, the efficient function of complex I and oxidative phosphorylation, the stimulation of K_ATP_ channels and the activation of hypoxic preconditioning mechanisms [27].

It has been demonstrated that ischemic preconditioning causes a very rapid increase in mCx43 levels [7,26], and early studies report that mCx43 plays an essential role in preventing apoptotic cell death [5]. Additionally, the opening of K_ATP_ channels has been reported as a part of the complex cytoprotective program activated during hypoxia [2,26]. The opening of K_ATP_ channels in cardiomyocytes improves the functional and energetic recovery of these cells after ischemic or hypoxic insults [16]. Mitochondrial K_ATP_ channel activation has also been shown to alter cardiac mitochondrial function, including the attenuation of mitochondrial Ca^2+^ overload as well as the release of membrane proteins such as cytochrome c [10]. 

This study reported that K_ATP_ is an interaction partner of Cx43, and here we used a cellular model of chemical-induced hypoxia to further elucidate the functional link between K_ATP_ channels and mCx43. 

Our results show that CoCl_2_ induces a remarkable increase in mCx43 expression. The opening of K_ATP_ channels exerted by dzx further increases mCx43, and increased levels of mCx43 phosphorylated on Ser368 have also been observed in dzx pre-treated cells. This data is in agreement with previous studies indicating that K_ATP_ channels interact with Cx43 in a phosphospecific manner [28] with a mechanism that involves protein kinase C activation [29]. Indeed, hypoxia-mediated translocation of PKCε to the mitochondria and its role in mediating cardioprotection are well established [30]. 

Several studies report that the increase in mCx43 expression and its phosphorylation is part of the cytoprotective mechanisms put in place by cells to counteract cellular damage induced by hypoxia [9,31] as well as by other cardiotoxic stimuli [8]. However, the increase in mCx43 expression alone appears to be ineffective in counteracting oxidative stress and mitochondrial damage induced by CoCl_2_. Indeed, an increase in mCx43 expression and in its phosphorylated form was also observed in the cells pre-treated with gly, the pharmacological tool used to close K_ATP_ channels, but our data showed further mitochondrial ROS production as well as mitochondrial membrane damage and apoptosis in these cells. On the contrary, in agreement with previous reports [17,32], the opening of K_ATP_ channels exerted by dzx administration not only increased mCx43 expression but also reduced mitochondrial ROS production, mitochondrial membrane depolarization and mitochondrial Ca^2+^ overload in CoCl_2_-treated cells, thus reducing hypoxia-induced apoptosis. Furthermore, all these cytoprotective effects of dzx disappear in cells in which the translocation of Cx43 to mitochondria was inhibited by radicicol administration, indicating that both mCx43 and K_ATP_ channels are required for cardioprotective effect. 

## 4. Materials and Methods

### 4.1. Materials

The cardiomyoblast cell line (H9c2) was purchased from the American Tissue Culture Collection (Manassas, VA, USA). Cobalt chloride (CoCl_2_), diazoxide (dzx), radicicol and glybenclamide (gly) were purchased from Sigma-Aldrich (Milan, Italy).

### 4.2. Cell Culture

The H9c2 embryonic rat heart-derived cells were subcultured weekly in 100 mm Corning dishes containing 10 mL of Dulbecco’s Modified Eagle Medium (DMEM; Gibco) with 10% *v*/*v* foetal bovine serum (FBS; Gibco, Rodano (MI), Italy), 100 U/mL of penicillin and 100 μg/mL of streptomycin at 37 °C in a humidified atmosphere of 5% CO_2_.

### 4.3. Experimental Protocols

Confluent H9c2 cells were treated for 3 h with CoCl_2_ (150 µM) in DMEM 10% FBS. Radicicol (1 µM) was used to inhibit Cx43 translocation to mitochondria, dzx (100 µM) and gly (100 µM) were used induce the opening or closure, respectively, of ATP-regulated potassium channels (K_ATP_). Dzx, gly or radicicol were added 30 min before the CoCl_2_ treatment and left in the incubation medium for the entire experiment time.

### 4.4. Mitochondrial Protein Extraction and Western Blot Analysis 

H9c2 cells (1.0 × 10^6^ cells/plate) were seeded into 100 mm Petri plates (Corning, Inc. New York, NY, USA) and treated as described above. Mitochondrial protein extraction was carried out from cells in lysis buffer A (K^+^ Hepes pH 7.5 20 mM, Sucrose 250 mM, KCl 10 mM, MgCl_2_ 1.5 mM, EGTA 1 mM, EDTA 1 mM, protease inhibitors, NaF 50 mM, Na_3_VO_4_ 0.2 mM, PMSF 1 mM, DTT 1 mM and digitonin 0.025%). Thereafter, the cells were centrifugated at 16,000× *g* for 2 min at 4 °C. The pellet was resuspended in lysis buffer B (NaDeOH 0.5%, Triton X 1%, Tris HCl 50 mM pH 7.4, NaCl 150 mM and SDS 1%) to obtain mitochondrial protein, after discarding the supernatant.

The mitochondrial protein concentrations were determined by the Bradford protein assay (Bio-Rad Laboratories, Inc. Hercules, CA, USA) using bovine serum albumin as standard. Equal amounts of protein (50 µg protein/lane) were loaded onto 10% SDS-PAGE under denaturing conditions. Nitroblots were probed with primary antibody anti-Cx43 (1:250; #3512, BD Transduction Laboratories) or anti-Tom20 used as a loading control (1:250; #sc-114115, Santa Cruz Biotechnology) overnight. After washing with TBS/0.1% Tween, the appropriate secondary antibody, anti-rabbit (#GTXRB-003-DHRPX) or anti-mouse (#GTXMU-003-DHRPX) (each diluted 1:4000), was incubated for 1 h at room temperature. Immunoreactive proteins were detected by the enhanced chemiluminescence reagents (ECL) and blot imaging (LAS 4000; GE Healthcare). The images were analysed for densitometry using ImageJ software. In order to verify the purity of the mitochondrial protein extraction, a Western blot analysis was performed to evaluate the absence of proteins expressed in other cellular compartments (Na^+^/K^+^-ATPase, #ab7671, Abcam, Cambridge, UK) and the presence of proteins expressed only in the mitochondria (ox-Phos Complex II, #ab14715, Abcam, Cambridge, UK) as previously reported [8,15].

### 4.5. Measurement of Mitochondrial ROS Production in H9c2 Cells

The mitochondrial superoxide formation was evaluated by MitoSOX Red (Molecular Probes, Invitrogen, #M36008 UK). This indicator is a fluorogenic dye for the highly selective detection of superoxide in the mitochondria of live cells as it is oxidized by superoxide but not by other ROS-generating systems, as previously reported [18,33,34]. For these experiments, H9c2 cells (4.0 × 10^5^ cells/well) were cultured in a 6-well plate and treated as described. After the incubation period, MitoSOX Red (2.5 μM) was added for 15 min at 37 °C, then cells were washed gently with PBS and collected for fluorescence evaluation by means of flow cytofluorometry. The cell fluorescence was evaluated using a FACS scan and analysed by CellQuest software.

### 4.6. Measurement of Mitochondrial Membrane Potential in H9c2 Cells

The mitochondrial membrane potential was monitored by tetramethylrhodamine ethyl ester (TMRE; Thermo Fisher, #T669 Invitrogen, Inchinnan, UK) fluorescence. TMRE is a fluorescent cationic dye that accumulates in the mitochondria in inverse proportion to the membrane potential; so, a low percentage of TMRE^+^ cells indicates that TMRE dye was not trapped in the mitochondrial membrane due to depolarization. Briefly, for these experiments H9c2 cells (4.0 × 10^5^ cells/well) were plated in a 6-well plate and treated as described above. After the incubation period, the cells were collected, washed twice with PBS and then incubated in PBS containing TMRE (5 nM) at 37 °C. After 30 min, the cell fluorescence was recorded by a FACS scan and evaluated with CellQuest software [9].

### 4.7. Evaluation of Intracellular Ca^2+^-Signalling

Changes in intracellular calcium concentration were measured using the calcium-sensitive fluorescent ratiometric dye Fura-2 AM (#F0888), the membrane-permeant acetoxymethyl ester form of Fura-2. This method is a high throughput way to measure agonist mediated calcium responses.

H9c2 cells (3 × 10^4^ cells/plate) were seeded in 86 mm tissue culture plates and treated as described above. After the incubation period, the H9c2 cells were washed in a phosphate buffer and resuspended in 1 mL of Hank’s balanced salt solution (HBSS) containing 5 µM Fura-2 AM for 45 min. After removing the Fura-2 AM excess, the cells were incubated in a Ca^2+^-free HBSS/0.5 mM EGTA buffer for 15 min to allow the hydrolysis of Fura-2 AM into its active-dye form, Fura-2. Then, the cells were analysed by a spectrofluorimeter (PerkinElmer LS-55). Ionomycin (1 µM final concentration, #I0634) or carbonyl cyanide p-trifluoromethoxy-phenylhydrazone (FCCP, 50 nM final concentration, # C2920), used to evaluate the content of calcium in sarcoendoplasmatic reticulum or in mitochondria, respectively, were added into the cuvette in a calcium-free HBSS/0.5 mM EGTA buffer. The excitation wavelength was alternated between 340 nm and 380 nm, and the emission fluorescence was recorded at 515 nm. The ratio of fluorescence intensity of 340 nm/380 nm (F340 nm/F380 nm) is strictly related to intracellular free calcium, as previously reported [35,36]. Data were expressed as percentage of the Δ increase of the fluorescence ratio (F340 nm/F380 nm) induced by 1 μmol/L ionomycin or 0.05 μmol/L FCCP–basal fluorescence/basal fluorescence ratio (F340 nm/F380 nm).

### 4.8. Measurement of Hypodiploid DNA Using Propidium Iodide Staining

The hypodiploid cells were identified through the use of a fluorochrome, propidium iodide (PI, #P4170), that is capable of binding cellular DNA content, and they were analysed by means of flow cytometry. H9c2 cells (4.0 × 10^5^ cells/well) were cultured in a 6-well plate and treated as described above. At the end of the treatment, the cells were washed in phosphate buffered saline (PBS) and resuspended in 500 μL of a solution containing 0.1% Triton X-100, 50 μg/mL of PI and 0.1% sodium citrate. After 30 min of incubation at 4 °C in the dark, the cell nuclei were analysed using CellQuest software. The results are expressed as the percentage of cells in the hypodiploid region.

### 4.9. Statistical Analysis

The results are expressed as the mean ± S.E.M. of at least three independent experiments, each performed in duplicate. Data were analysed by one way ANOVA and multiple comparisons were made using the Bonferroni post-test. A value of *p* < 0.05 was considered as statistically significant.

## 5. Conclusions

In conclusion, here we highlighted the cooperative interaction between mCx43 and K_ATP_. In fact, both the inhibition of Cx43 translocation to mitochondria and the closure of K_ATP_ channels drastically affect mitochondrial homeostasis, thus confirming that the interaction of Cx43 and K_ATP_ augments each other’s functions. The main limitation of this study is that it is focused on the functional link between mCx43 and K_ATP_ channels. Despite our results supporting previous observations regarding the cardioprotective effects of diazoxide in the presence of functional mCx43, further studies are needed to better clarify the pathways activated and to identify novel therapeutic applications for this drug. 

## Figures and Tables

**Figure 1 ijms-22-11599-f001:**
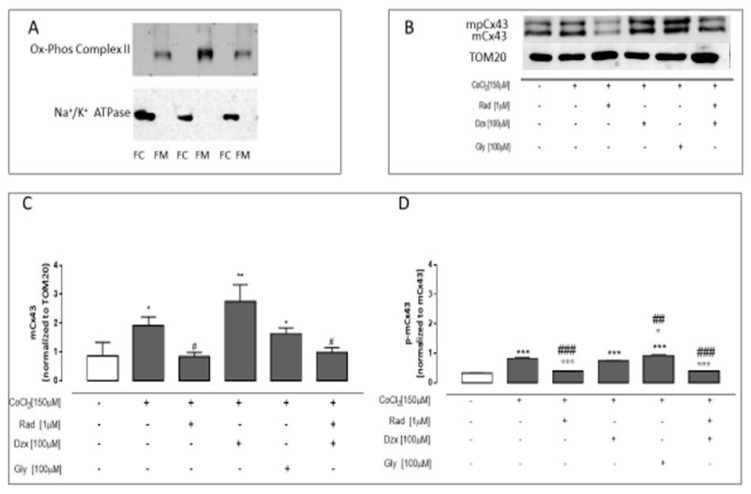
Effects of the opening of K_ATP_ channels on mitochondrial Cx43 expression in CoCl_2_-induced hypoxia. H9c2 cells were treated for 3 h with CoCl_2_ (150 µM); rad (1 µM), dzx (100 µM) and gly (100 µM) were added 30 min before CoCl_2_ administration. Panel (**A**) shows representative Western blots of Na^+^/K^+^-ATPase and of OXPHOS Complex II used as markers to demonstrate the purity of the cytosolic and mitochondrial extracts, respectively. Mitochondrial Cx43 expression was detected by Western blotting. Tom20 protein expression was used as loading control (**B**). Panel (**C**) shows the histograms of mCx43 normalized to Tom20. Panel (**D**) shows the histograms of mpCx43 normalized to mCx43. Results are expressed as mean ± S.E.M. from at least three independent experiments, each performed in duplicate. Data were analysed by one way ANOVA and multiple comparisons were made by Bonferroni post-test. * *p* < 0.05, ** *p* < 0.005 and *** *p* < 0.001 vs. untreated cells; ° *p* < 0.05 and °°° *p* < 0.001 vs. CoCl_2_-treated cell; # *p* < 0.05, ## *p* < 0.005 and ### *p* < 0.001 vs. CoCl_2_ and dzx co-treated cell.

**Figure 2 ijms-22-11599-f002:**
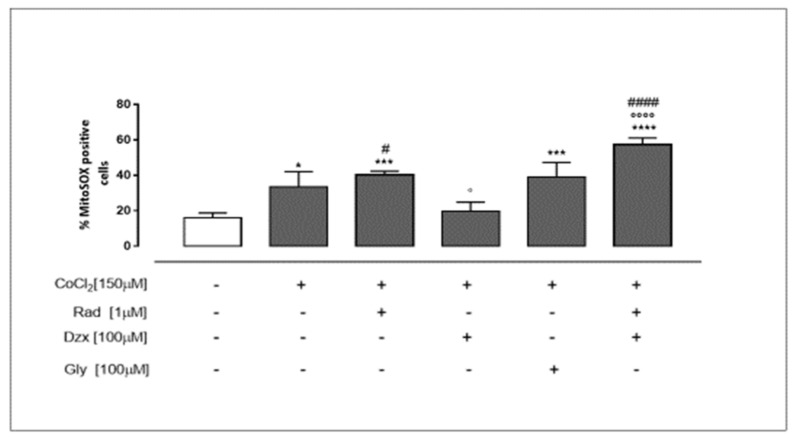
Mitochondrial Cx43 and mitochondrial K_ATP_ channels are involved in the reduction of CoCl_2_-induced superoxide production. H9c2 cells were treated for 3 h with CoCl_2_ (150 µM); rad (1 µM), dzx (100 µM) and gly (100 µM) were added 30 min before CoCl_2_ administration. Mitochondrial superoxide production was evaluated by means of the probe MitoSOX Red by flow cytometry analysis. Results are expressed as mean ± S.E.M. of percentage of MitoSOX positive cells of at least three independent experiments, each performed in duplicate. Data were analysed by Bonferroni post-test. * *p* < 0.05,*** *p* < 0.001 and **** *p* < 0.0001 vs. untreated cells; ° *p* < 0.05 and °°°° *p* < 0.0001 vs. CoCl_2_-treated cell; # *p* < 0.05 and #### *p* < 0.0001 vs. CoCl_2_ and dzx co-treated cell.

**Figure 3 ijms-22-11599-f003:**
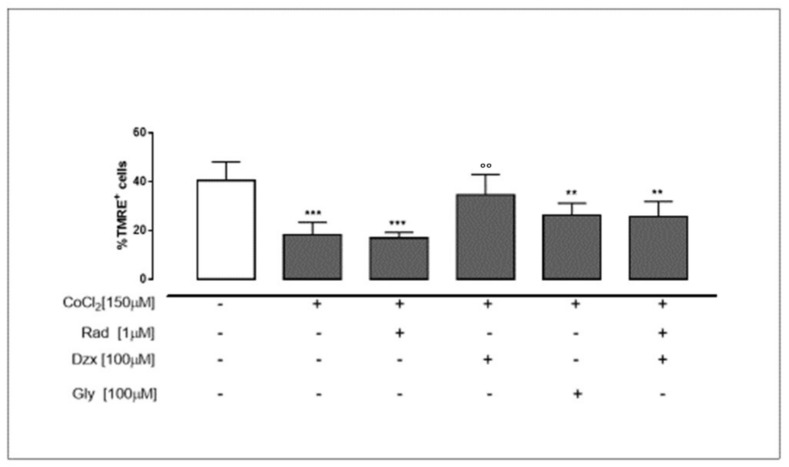
Mitochondrial Cx43 and the opening of mitochondrial K_ATP_ channels prevent mitochondrial membrane depolarization. Cells were treated for 3 h with CoCl_2_ (150 µM); rad (1 µM), dzx (100 µM) and gly (100 µM) were added 30 min before CoCl_2_ administration. Flow cytometry analysis was carried out through tetramethylrhodamine ethyl ester (TMRE), a red–orange positive dye that penetrates and accumulates in the mitochondria in inverse proportion to the membrane potential, highlighting the mitochondrial membrane potential. The low value of the percentage of TMRE^+^ cells implies that the TMRE dye was not caught in the mitochondrial membrane due to its depolarization. Results were expressed as mean ± S.E.M. of fluorescence intensity of at least three independent experiments, each performed in duplicate. Data were analysed by one way ANOVA and multiple comparisons were made by Bonferroni post-test. ** *p* < 0.005 and *** *p* < 0.001 vs. untreated cells; °° *p* < 0.005 vs. CoCl_2_-treated cell.

**Figure 4 ijms-22-11599-f004:**
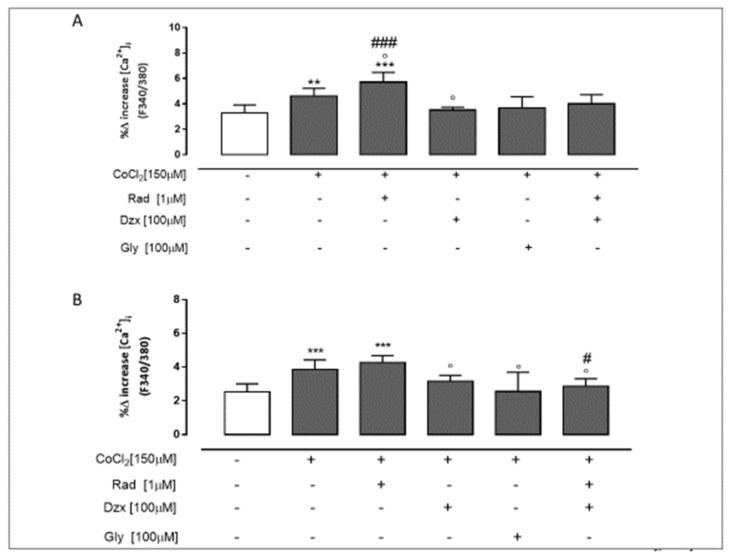
Mitochondrial Cx43 and the opening of mitochondrial K_ATP_ channels prevent CoCl_2_-induced Ca^2+^ homeostasis alteration. Cells were treated for 3 h with CoCl_2_ (150 µM); rad (1 µM), dzx (100 µM) and gly (100 µM) were added 30 min before CoCl_2_ administration. Panel (**A**) show the intracellular calcium content valued on H9c2 cells in calcium-free medium by means of ionomycin (1 μM). Mitochondrial calcium pool was detected on H9c2 cells in calcium-free medium in presence of FCCP (50 nM) (panel (**B**)). Data were expressed as mean ± S.E.M. of delta (δ) increase in FURA-2 ratio fluorescence (340/380 nm) from at least three independent experiments, each performed in triplicate. Data were analysed by one way ANOVA and multiple comparisons were made by Bonferroni post-test. ** *p* < 0.005 and *** *p* < 0.001 vs. untreated; ° *p* < 0.05 vs. CoCl_2_-treated cell; # *p* < 0.05 and ### *p* < 0.001 vs. CoCl_2_ and dzx co-treated cell.

**Figure 5 ijms-22-11599-f005:**
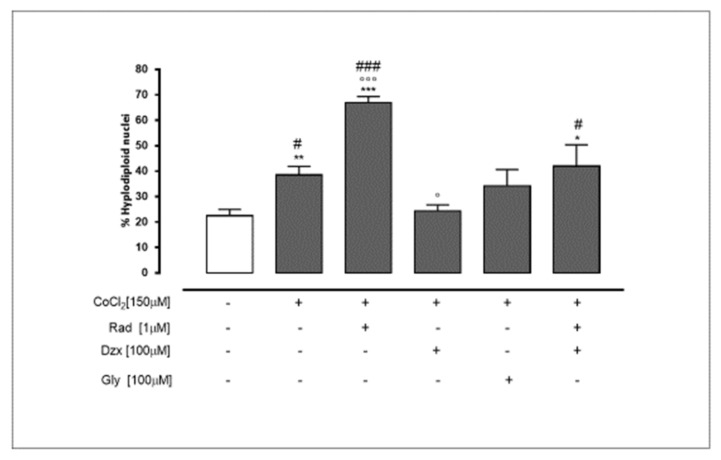
Mitochondrial Cx43 and the opening of mitochondrial K_ATP_ channels counteract CoCl_2_-induced apoptotic response. Cells were treated for 3 h with CoCl_2_ (150 µM); rad (1 µM), dzx (100 µM) and gly (100 µM) were added 30 min before CoCl_2_ administration. H9c2 were stained by propidium iodide and fluorescence of individual nuclei was measured by flow cytometry. Results are expressed as mean ± S.E.M. of percentage of hypodiploid nuclei from at least three independent experiments, each performed in duplicate. Data were analysed by one way ANOVA and multiple comparisons were made by Bonferroni post-test. * *p* < 0.05, ** *p* < 0.005 and *** *p* < 0.001 vs. untreated cells; ° *p* < 0.05 and °°° *p* < 0.001 CoCl_2_-treated cell; # *p* < 0.05 and ### *p* < 0.001 vs. CoCl_2_ and dzx co-treated cell.

## Data Availability

The data presented in this study are available on request from the corresponding author.

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
