# Peer review of "Diazoxide Needs Mitochondrial Connexin43 to Exert Its Cytoprotective Effect in a Cellular Model of CoCl2-Induced Hypoxia"

_ijms, 2021, doi:10.3390/ijms222111599_

Round 1
Reviewer 1 Report
- The quality of the illustrations needs to be improved
- MitoSOX is shown to be distriburted in mitochondria but also in cytosol. Consider this the image of MitoSOX with points of measurements should be presented
- Fluorescent Ca2+ measurement is an essential assay for this manuscript. Authors need to indicate the linearity of this assay for the experiments and how this has been optimized. The data can be included as a supplementary.
- The introduction and discussion should discuss the effects of Cx43 on the regulation of Ca2 + homeostasis of normal and cancer cells under various influences. For example: https://pubmed.ncbi.nlm.nih.gov/34439975/, https://pubmed.ncbi.nlm.nih.gov/34360859/
Author Response
REVIEWER 1
1. The quality of the illustrations needs to be improved
Reply. As suggested we improved the quality of the illustrations. In particular, we inserted the figures
with an higher resolution (300dpi).
2. MitoSOX is shown to be distriburted in mitochondria but also in cytosol. Consider this the image of
MitoSOX with points of measurements should be presented
Reply. We agree with the reviewer’s observation, The selectivity of MitoSOX distribution in
cytosol/mithocondria is controversial even if it seems related to the experimental conditions.
This aspect is discussed by Kauffman and co-workers (MitoSOX-Based Flow Cytometry for Detecting
Mitochondrial ROS. React Oxyg Species (Apex) 2016;2(5):361-370. doi: 10.20455/ros.2016.865; also
included in the manuscript). They report that MitoSOX distribution in cytosol/mithocondria is related to the
experimental conditions. As they stated “MitoSOX is positively charged probe that rapidly accumulates in
mitochondria, and as such may be used to detect superoxide/ROS production inside mitochondria via
fluorometry, microscopy, or flow cytometry. In fact, fluorescence imaging of the dihydroethidium/MitoSOXstained cells or tissues has been claimed as a selective assay for intracellular and intra-mitochondrial
superoxide production, but this claim has received criticism”.
Nevertheless, they concluded that measurement of MitoSOX-derived fluorescence intensity, when the
probe is used at appropriate concentrations (2.5 µM, as in our experiments), seems to be reflective of the
levels of mitochondrial total ROS.
Our data are reported as percentage of MitoSOX positive cells, considering that the percentage is referred
at our "gate", which is the sequential identification and refinement of our cellular population using MitoSOX
Red as a marker that is visualized by fluorescence in a unique emission spectrum according previous
studies (e.g. Wojtala A, et al. Methods Enzymol. 2014; 542:243-62. doi: 10.1016/B978-0-12-416618-
9.00013-3; Yang Y, et al.,STAR Protoc. 2021 Apr 20;2(2):100466. doi: 10.1016/j.xpro.2021.100466;
Pecoraro M, et al., Toxicol In Vitro 2020, 67: 104926; Pecoraro M, et al., Toxicol In Vitro 2018, 47:120-128;
Pecoraro M, et al., Cardiovasc Toxicol. 2015 Oct;15(4):366-76. doi: 10.1007/s12012-014-9305-8).
3. Fluorescent Ca2+ measurement is an essential assay for this manuscript. Authors need to
indicate the linearity of this assay for the experiments and how this has been optimized. The data
can be included as a supplementary
Reply. The measurement of intracellular Ca2+ signaling was performed by spectrofluorimetry using the
probe Fura 2-AM (Sigma) in agreement with the method reported by Improta-Brears with minor revisions.
(T Improta-Brears , et al., Proc Natl Acad Sci U S A 1999 Apr 13; 96 (8): 4686-91. doi:
10.1073/pnas.96.8.4686.
Regarding the linearity of this assay for the experiments and how this has been optimized we proceeded
as follow: For each sample, mitochondrial calcium content was evaluated as follow: [(F340/F380 nm after
FCCP administration - F340/F380 nm basal) / F340/F380 nm basal] x 100 and expressed as % Δ increase
of Fura 2 ratio fluorescence (F340/F380 nm). Cytosolic calcium content was evaluated as: [(F340/F380 nm
after Ionomicyn administration - F340/F380 nm basal) / F340/F380 nm basal] x 100 and expressed as % Δ
increase of Fura 2 ratio fluorescence (F340/F380 nm).
Data on graph report mean ± SEM of % Δ increase from three independent experiments each performed
in triplicate.
More specifically, after treatments, cells were loaded with the probe suspended in Ca2+
-free Hanks
balancced salt solution at a final concentration of 5 µM at 37 °C for 45 min with a subsequent 15 min
washout. Then cells were transferred in quarz cuvette and analysed at the spectrofluorimeter. During the
analysis the cells were kept stirred with a magnet. Basal Ca2+ level (F340/F380 nm ratio) was acquired after
200 s of stabilization, then FCCP (50 nM final concentration) was added and the F340/F380 nm ratio was
acquired. Ionomicyn (1 µM final concentration) was added 200s after FCCP administration and the F340-
/F380 nm ratio was acquired.
4. The introduction and discussion should discuss the effects of Cx43 on the regulation of Ca2 +
homeostasis of normal and cancer cells under various influences. For example:
https://pubmed.ncbi.nlm.nih.gov/34439975/, https://pubmed.ncbi.nlm.nih.gov/34360859/
Reply. Thank you for your input. We provided to discuss Cx43 on the regulation of Ca2 + homeostasis
of normal and cancer cells under various influences in the Introduction section (lines 40-41) and in
discussion section (lines 316-321) also considering the suggested references.
Reviewer 2 Report
In this article, “Diazoxide needs mitochondrial Connexin43 to exert its cytoprotective effect in a cellular model of CoCl2-induced hypoxia”, set out to study the cytoprotective effects of Diazoxide during hypoxia in cardiac myocytes.
The topic of this manuscript is, in general, quite interesting. The manuscript is well written. There are some major issues that need to be addressed.
1-One major issue is the use of CoCl2. Why do the authors use CoCl2 instead of hypoxia (hypoxia chamber). Why chemical hypoxia? I suggest at least adding a positive control group to their experiments using low oxygen levels (hypoxia chamber) for better comparison.
2- I think addition of a schematic figure will help to better understand the discussed items in the manuscript.
3-Do the authors think that their results are affected by viability of cells in culture using these treatments? In another word do the treatments used in their experiments affect cellular viability. If so, to what extent. Do they have data to support that?
4- Cobalt chloride imitates hypoxia in vitro by stabilizing hypoxia-inducible factor 1- alpha which upregulates certain cytoprotective and cytodestructive downstream genes. It is worthwhile discussing the interaction between HIF and Connexin43 and the link between them. The authors may use these as references:
DOI: 10.3892/mmr.2020.10966
https://doi.org/10.1007/s11010-021-04082-9
5-Do the authors have better loading control for their western blot experiment? It appears that in panel B there’s un-equal loading based on the blot.
6-Why did the authors not evaluate mRNA upregulation of the studied proteins?
7-Please add a section regarding the limitation of the study, future direction, and possible translational and therapeutic usages for their findings.
Round 2
Reviewer 1 Report
The authors responded to my comments. I am recommend the article for publication.
Reviewer 2 Report
The authors adequately addressed the issues and the manuscript improved substantially. I have no further suggestions. Thanks.